# DNA methylation in newborns conceived by assisted reproductive technology

Siri E. Håberg[1✉], Christian M. Page[1,2], Yunsung Lee [1], Haakon E. Nustad[1,3], Maria C. Magnus[1,4,5], Kristine L. Haftorn[1], Ellen Ø. Carlsen[1], William R. P. Denault[1,6], Jon Bohlin [1,7], Astanand Jugessur[1,8], Per Magnus[1], Håkon K. Gjessing[1,8] & Robert Lyle[1,9]

Assisted reproductive technology (ART) may affect fetal development through epigenetic mechanisms as the timing of ART procedures coincides with the extensive epigenetic remodeling occurring between fertilization and embryo implantation. However, it is unknown to what extent ART procedures alter the fetal epigenome. Underlying parental characteristics and subfertility may also play a role. Here we identify differences in cord blood DNA methylation, measured using the Illumina EPIC platform, between 962 ART conceived and 983 naturally conceived singleton newborns. We show that ART conceived newborns display widespread differences in DNA methylation, and overall less methylation across the genome. There were 607 genome-wide differentially methylated CpGs. We find differences in 176 known genes, including genes related to growth, neurodevelopment, and other health outcomes that have been associated with ART. Both fresh and frozen embryo transfer show DNA methylation differences. Associations persist after controlling for parents' DNA methylation, and are not explained by parental subfertility.

[1] Centre for Fertility and Health, Norwegian Institute of Public Health, P.O. box 222 Skøyen, 0213 Oslo, Norway. [2] Department of Mathematics, Faculty of Mathematics and Natural Sciences, University of Oslo, P.O box 1032 Blindern, N-0315 Oslo, Norway. [3] Deepinsight, Karl Johans gate 8, 0154 Oslo, Norway. [4] MRC Integrative Epidemiology Unit at the University of Bristol, Oakfield House, Oakfield Grove, Bristol BS8 2BN, UK. [5] Population Health Sciences, Bristol Medical School, Oakfield House, Oakfield Grove, Bristol BS8 2BN, UK. [6] Department of Human Genetics, University of Chicago, 5801S Ellis Ave, Chicago, IL 60637, USA. [7] Department of Method Development and Analytics, Norwegian Institute of Public Health, P.O. box 222 Skøyen, 0213 Oslo, Norway. [8] Department of Global Public Health and Primary Care, University of Bergen, P.O. box 7804, N-5020 Bergen, Norway. [9] Department of Medical Genetics, Oslo University Hospital, OUS HF, P.O. box 4956 Nydalen, 0424 Oslo, Norway. ✉email: Siri.haberg@fhi.no

ssisted reproductive technology (ART) use is increasing, and around 8 million children have been conceived using ART[1]. As more women postpone childbearing, and egg freezing is becoming more socially acceptable, this increase is expected to continue. ART is associated with several adverse pregnancy outcomes[2], with implications for childhood and adult health[3,4]. The impact of ART on long-term health is not clear[5]. Some studies have suggested an influence on neurodevelopment, cardiovascular function, metabolism, growth, and malignancies[4]. However, it is unclear whether the differences observed in ART-conceived children are caused by the ART procedure itself or by underlying factors associated with parental subfertility[6–8].

Epigenetic mechanisms regulate gene activity, cell function, and development. DNA methylation, the attachment of a methyl group to a cytosine preceding a guanine base (CpG), is an epigenetic mechanism essential for normal embryonic development[9]. ART involves the manipulation and culturing of embryos during a period that coincides with extensive epigenetic remodeling[9]. It is therefore plausible that ART procedures alter the fetal epigenome[9,10], potentially perturbing development[10,11] and thereby influencing later health. So far, there is limited and inconsistent evidence for ART-induced DNA methylation changes[12–22], with the exception of persistent findings for rare imprinting disorders[23,24].

In this work, we compare cord blood DNA methylation in ART conceived to naturally conceived newborns in a subsample of the Norwegian Mother, Father and Child Cohort study (MoBa)[25]. We also explore whether any differences are associated with ART methods. To address the potential influence of underlying parental subfertility, we control for DNA methylation in the parents, and test for differences in DNA methylation related to subfertility.

Here we show that ART conceived newborns display widespread differences in DNA methylation, with overall lower methylation across the genome, and 607 genome-wide differentially methylated CpGs. We find differences in 176 known genes, including genes related to growth, neurodevelopment, and other health outcomes that have been associated with ART. Both fresh and frozen embryo transfer show DNA methylation differences. Associations persist after controlling for parents' DNA methylation and are not explained by parental subfertility.

## Results

After data processing and quality control, DNA methylation data generated using the Illumina MethylationEPIC array was available for 962 ART conceived and 983 naturally conceived newborns (Table 1 and Fig. 1), and for 1,956 mothers and 1,949 fathers, which comprised 1,917 complete trios. DNA methylation measurement of 770,586 autosomal CpGs was available for each individual in the final data set (see online "Methods" for details).

**DNA methylation in ART versus naturally conceived newborns**. Newborns conceived by ART exhibited genome-wide slightly lower methylation (Fig. 2a). Overall, 74% of the CpGs were hypomethylated and 26% were hypermethylated in ART conceived newborns. No such shift was observed when comparing DNA methylation levels in the parents of these children (Fig. 2b, c). The distribution of the effect sizes in children was shifted to the left (median value −17.8E−03, 95% confidence interval (CI) −17.9E−03, −17.7E−03), while the distributions of the effect sizes in parents were rather symmetrical around zero: median value for mothers −0.6E−03 (95% CI −0.7E−03, −0.5E−03), and median value for fathers 4.7E−03 (95% CI 4.6E−03, 4.8E−03). This global shift in DNA methylation was seen in all genomic features (Supplementary Fig. 1). We identified 607 differentially methylated CpGs between ART and naturally conceived newborns at a false

**Table 1 Characteristics of the study participants.**

| Characteristics | Naturally conceived (n = 983) | ART conceived (n = 962) | P value[a] |
|---|---|---|---|
| Maternal age at delivery, mean (SD) | 30.0 (4.6) | 33.2 (3.6) | <0.001 |
| Paternal age at delivery, mean (SD) | 32.6 (5.4) | 35.8 (5.4) | <0.001 |
| Maternal parity, No. (%) | | | <0.001 |
| Nulliparous | 461 (46.9) | 673 (70.0) | |
| Multiparous | 522 (53.1) | 289 (30.0) | |
| Maternal educational level, No. (%) | | | 0.001 |
| Less than high school | 71 (7.2) | 48 (5.0) | |
| High school | 299 (30.4) | 232 (24.1) | |
| Up to 4 years of college | 386 (39.3) | 418 (43.5) | |
| More than 4 years of college | 225 (22.9) | 258 (26.8) | |
| Missing | 2 (0.2) | 6 (0.6) | |
| Paternal educational level, No. (%) | | | 0.004 |
| Less than high school | 98 (10.0) | 72 (7.5) | |
| High school | 398 (40.5) | 333 (34.6) | |
| Up to 4 years of college | 249 (25.3) | 296 (30.8) | |
| More than 4 years of college | 209 (21.3) | 235 (24.4) | |
| Missing | 29 (3.0) | 26 (2.7) | |
| Maternal pre-pregnancy BMI, mean (SD) | 24.3 (4.5) | 24.3 (4.0) | 0.572 |
| Missing, No. (%) | 14 (1.4) | 17 (1.8) | |
| Paternal pre-pregnancy BMI, mean (SD) | 25.9 (3.2) | 26.4 (3.5) | 0.002 |
| Missing, No. (%) | 23 (2.3) | 20 (2.1) | |
| Maternal smoking during pregnancy, No. (%) | | | <0.001 |
| Never | 490 (49.8) | 494 (51.4) | |
| Former | 253 (25.7) | 358 (37.2) | |
| Quit before 18 gestational weeks | 132 (13.4) | 62 (6.4) | |
| Continued after 18 gestational weeks | 104 (10.6) | 44 (4.6) | |
| Missing | 4 (0.6) | 4 (0.6) | |
| Paternal smoking, No. (%) | | | 0.459 |
| No | 747 (76.0) | 737 (76.6) | |
| Yes | 234 (23.8) | 220 (22.9) | |
| Missing | 2 (0.2) | 5 (0.5) | |
| Child sex, No. (%) | | | 0.073 |
| Male | 470 (47.8) | 505 (52.5) | |
| Female | 513 (52.2) | 457 (47.5) | |
| Child gestational age, mean (SD) | 39.6 (1.6) | 39.4 (1.7) | 0.210 |
| Missing, No. (%) | 4 (0.4) | 0 (0) | |
| Child birthweight, mean (SD) | 3649 (526) | 3526 (539) | <0.001 |
| Missing, No. (%) | 0 (0) | 1 (0.1) | |
| Time to pregnancy, No. (%) | | | |
| No information | 179 (18.2) | NA | |
| <3 months | 417 (42.4) | NA | |
| 3–8 months | 256 (26.0) | NA | |
| 9–12 months | 52 (5.3) | NA | |
| >12 months | 79 (8.0) | NA | |
| Type of ART, No. (%) | | | |
| Fresh embryo transfer IVF | NA | 437 (45.4) | |
| Fresh embryo transfer ICSI | NA | 327 (34.0) | |
| Frozen embryo transfer IVF | NA | 87 (9.0) | |
| Frozen embryo transfer ICSI | NA | 39 (4.1) | |
| Combination/unspecified | NA | 72 (7.5) | |

SD standard deviation, BMI body mass index, IVF in vitro fertilization, ICSI intracytoplasmic sperm injection.
[a]P values from one sided chi-square tests (categorical variables) or two-sided t-test (continuous variables).

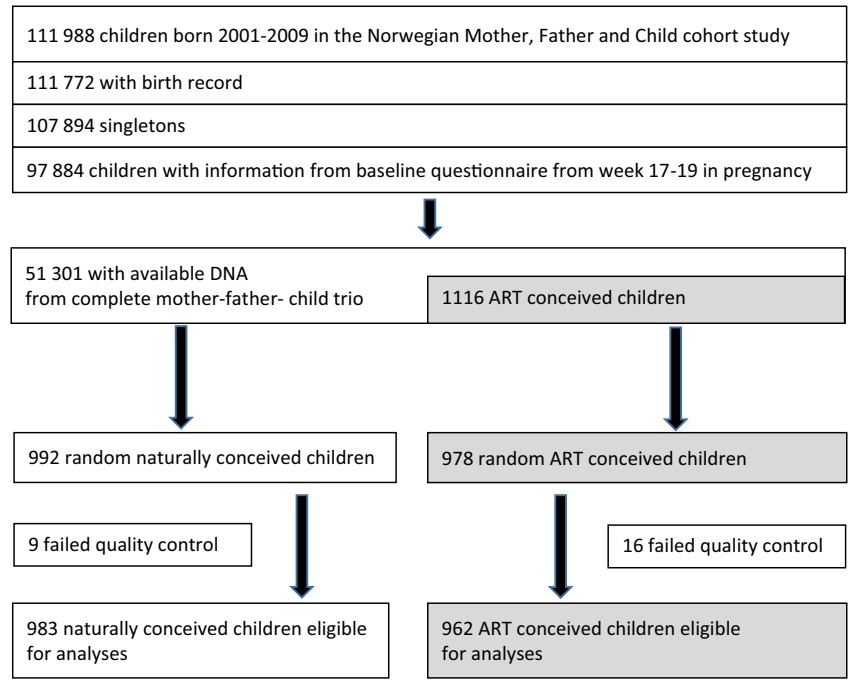

**Fig. 1 Selection of study participants.** The flow chart shows the selection of the eligible study population with 983 naturally conceived newborns and 962 ART conceived newborns in Norwegian Mother, Father and Child Cohort Study.

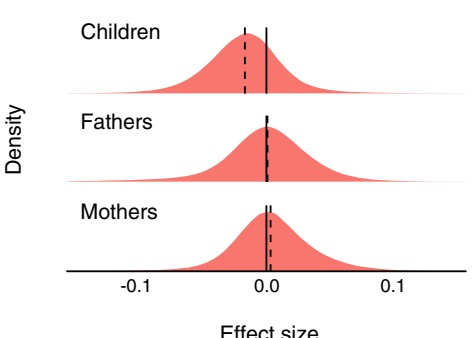

**Fig. 2 Distribution of the differences in DNA methylation between ART conceived and naturally conceived.** The effect sizes on the x-axis refer to the coefficients from the regression of DNA methylation at each CpG site on ART conceived versus naturally conceived, for newborns, fathers, and mothers. The vertical dashed line indicates the median value of the effect sizes. The adjusting variables for newborns were maternal age, maternal smoking, maternal BMI before pregnancy, child sex, parity, and plate ID. The adjusting variables for fathers were paternal age, paternal smoking, and paternal BMI. The adjusting variables for mothers were maternal age, maternal smoking, maternal BMI before pregnancy, and parity. Source data are provided as a Source Data file.

discovery rate (FDR) less than 0.01 (Fig. 3a, b, and Supplementary Data 1). Of the 607 significant CpGs identified after adjusting for maternal age, maternal smoking, maternal BMI, child sex, parity, and plate ID, 597 (98%) remained statistically significant after additional adjustment for gestational age. Further adjustment for birthweight and cord blood cell composition yielded similar results (Supplementary Fig. 2). After additional adjustment for maternal educational level and paternal age, 520 (86%) CpGs remained statistically significant (Fig. 3c). There were negligible differences

in cord blood cell composition between ART and naturally conceived newborns (Supplementary Fig. 3).

Although the overall pattern was hypomethylation in ART conceived children, for the 607 significant CpGs, 35% were hypomethylated and 65% were hypermethylated in ART conceived newborns. The two first principal components of the 607 CpGs showed a shift in newborns conceived with ART, with two distinct clusters for both sexes, that was not observed in the parents (Fig. 4).

**Different ART methods and newborn DNA methylation.** When comparing newborns conceived using fresh embryo transfer ($n = 764$) to naturally conceived newborns ($n = 983$), the number of significant CpGs increased to 800 (Fig. 5a). As the majority (77%) of the ART conceived newborns in our study were conceived using fresh embryo transfer, there was a considerable overlap (524 of 607 CpGs, 86%) with the CpGs found when comparing all ART to naturally conceived. When frozen embryo transfer ($n = 126$) was compared to naturally conceived, there were three statistically significant CpGs at FDR < 0.01, (Fig. 5b and Supplementary Table 1). These were also significant in the fresh vs natural comparison. There were three other statistically significant CpGs when comparing frozen to fresh embryo transfer (Fig. 5c and Supplementary Table 1). When compared to naturally conceived, there were 102 significant CpGs for in vitro fertilization (IVF) without intracytoplasmic sperm injection (ICSI) ($n = 511$) (Fig. 5d), and 55 CpGs for IVF with ICSI ($n = 359$) (Fig. 5e), of which 24 CpGs overlapped for both methods. We found no significant differences in DNA methylation levels when comparing newborns conceived with ICSI directly to those conceived without the use of ICSI (Fig. 5f).

**Parental subfertility and newborn DNA methylation.** The observed difference between ART and naturally conceived newborns persisted after adjustment for parental methylation levels (Fig. 6).

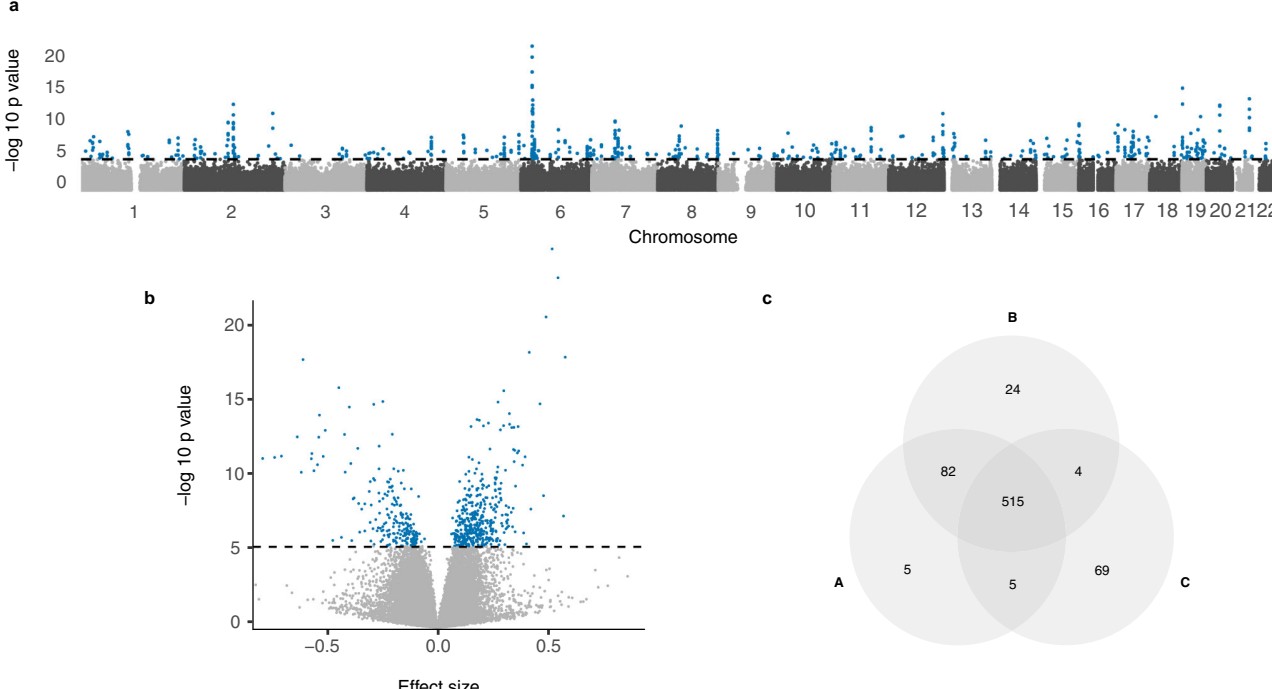

**Fig. 3 Associations with ART and epigenome-wide DNA methylation in newborns.** The *P* values (calculated from *t* statistics) were derived from two-sided *t* tests for comparing the DNA methylation levels between ART conceived (*n* = 962) and naturally conceived newborns (*n* = 983), with adjustment for maternal age, maternal smoking, maternal BMI before pregnancy, child sex, parity, and plate ID. **a** Manhattan plot with −log$_{10}$ *P* values plotted against the genomic position. **b** Volcano plot displaying the magnitude of the effect sizes against their corresponding −log$_{10}$ *P* value. The horizontal dashed line indicates FDR < 0.01. Blue dots indicate significant CpGs at FDR < 0.01, while gray dots indicate nonsignificant CpGs. The dashed lines represent a genome-wide significance cut-off of FDR < 0.01. **c** Overlap of significant CpGs in association with ART from a Wald test corrected for multiple hypotheses (FDR < 0.01) for three different models. A: adjusted for maternal age, maternal smoking, maternal BMI, child sex, parity, and plate ID. Same models as (**a**) and (**b**). B: adjusted for the covariates in A and gestational age at birth. C: adjusted for the covariates in A and gestational age at birth, birthweight, maternal educational level, paternal age, and cord blood cell composition. Source data are provided as a Source Data file.

There was no evidence of a difference in newborns' DNA methylation with increasing time to pregnancy (TTP) (Supplementary Fig. 4). Similarly, there were no significant CpGs when comparing those conceived after a TTP of more than 12 months (*n* = 79) to those conceived within three months (*n* = 417).

**Newborn DNA methylation and potential health implications.** The CpGs are annotated to 176 genes, 55 of which contained at least two significant CpGs (Table 2). Mutations in 14 of these genes cause Mendelian disorders, nine of them with a neurological phenotype. Based on data from mouse targeted mutagenesis, at least four genes are involved in fertility (Supplementary Data 2). The distribution of the methylation level at differentially methylated regions for the eight genes with the highest number (eight or more) of significant CpGs between ART and naturally conceived newborns are shown in Supplementary Fig. 5. One gene had 11 differentially methylated CpGs: the *HLA-DQB2* (Major Histocompatibility Complex, Class II, DQ Beta 2), which belongs to the human leukocyte antigen (HLA) complex. There were 10 differentially methylated CpGs within the bidirectional promoter of *BRCA1* and *NBR2* in newborns conceived by ART (Fig. 7). The parents using ART did not have similar DNA methylation differences when compared to parents conceiving without ART (Fig. 7). CpGs previously found to be significantly associated with birthweight[26] and Mendelian neurodevelopmental disorders[27] were overrepresented among the 607 significant ART associated CpGs in our data (Fig. 8). There were five genes overlapping

with results from other studies, and the direction of methylation differences in these studies was consistent with our results (Supplementary Table 2).

**Discussion**
We observed differences in DNA methylation at birth in 607 CpGs when comparing ART-conceived to naturally conceived newborns. These differences were not explained by parents' methylation, and not seen with underlying subfertility. There were differences in genes related to growth and disease, suggesting potential implications for development and health. There was a global shift toward lower methylation in newborns conceived with ART, and a similar shift was not seen when comparing ART parents to non-ART parents. Despite overall less methylation across the genome in newborns conceived with ART, most individual CpGs (65%) among the 607 top hits were hypermethylated. The potential health implications of a global shift towards lower methylation are not known.

We found several differentially methylated CpGs within genes associated with Mendelian disorders and fertility. There were two genes with 10 and 11 significant CpGs, *BRCA1* and *HLA-DQB2*, respectively. *BRCA1* plays a pivotal role in genome maintenance, cell division, and gene expression[28,29], and is a susceptibility gene for early-onset breast cancer[30]. *HLA-DQB2* is part of the HLA system, a group of related proteins encoded by the MHC involved in both normal immune response and disease pathogenesis[31]. HLA-DQB2 is an MHC class II protein, which presents antigens from outside of the cell to T-lymphocytes, however very little is

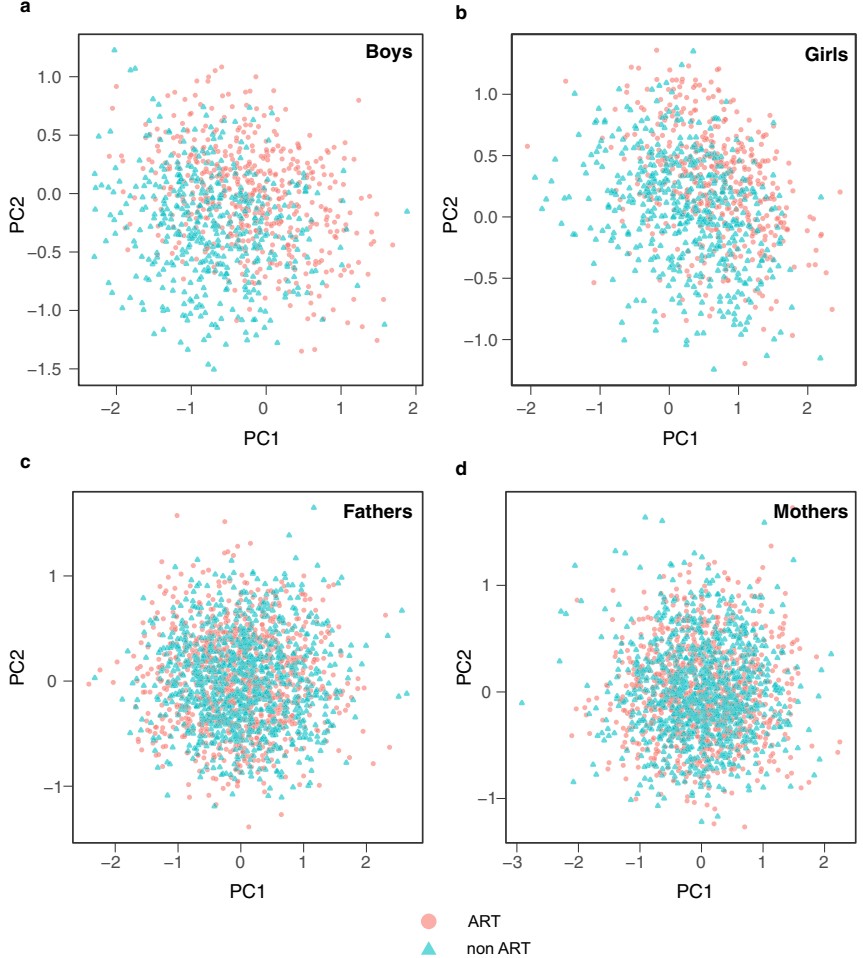

**Fig. 4 Principal component analysis of DNA methylation data in ART conceived and naturally conceived newborns, and their parents.** Scatter plots display the first two principal components (PC1 and PC2) of ART conceived (pink circles) versus naturally conceived (blue triangles) boys and girls, and their parents. **a** 505 ART conceived boys and 470 naturally conceived boys were examined. **b** 457 ART conceived girls and 513 naturally conceived girls were examined. **c** 969 fathers for ART conceived children, 980 fathers for naturally conceived children were examined. **d** 970 mothers for ART conceived children, 986 mothers for naturally conceived children were examined. Source data are provided as a Source Data file.

known about its function. Replication of these findings and gene expression studies are required to determine if altered DNA methylation is correlated with gene expression, and whether these differences in DNA methylation in newborns have potential implications on later health.

Birthweight has been shown to differ consistently in children conceived using ART.

Birthweight has also been associated with differentially methylated CpGs in newborns[32]. We found a strong enrichment for birthweight-related CpGs, suggesting a connection between ART, DNA methylation, and birthweight. Some neurodevelopmental outcomes have been suggested to differ in children conceived using ART, although results are inconsistent[5]. We found an enrichment of CpGs associated with neurodevelopment[27] in ART-conceived children, indicating a potential relationship between ART and methylation of CpGs in genes involved in neurodevelopment.

Although a number of studies have looked at the connection between DNA methylation and ART, the results have been inconsistent due to differences in sample selection, sample size, ART treatments, and analysis methodology (DNA source, experimental DNA methylation measurement, cell composition adjustment, covariates, statistical analyses, and significance testing)[12–22]. However, two recent publications have some overlap with our findings. Novakovic et al.[14] identified 18 differentially methylated regions (DMR) annotated to genes, two of which are also significant in our results (*CHRNE* and *NECAB3*). Yeung et al.[12] identified 10 genes, four are among our results (*MYO1D*, *GET1*, *NECAB3*, *KLK4*). The gene *NECAB3* was identified in all three studies, and three of the five overlapping genes are represented by multiple CpGs, with the same direction of DNA methylation differences for all CpGs.

Our study is unique in terms of the large sample size. This enabled us to evaluate different ART procedures and control for potential confounders, using a stringent threshold for declaring statistical significance. Moreover, DNA methylation measurements on complete mother-father-child trios made it possible to rule out parental methylation and subfertility as explanations for the differences observed in the ART conceived versus naturally conceived newborns.

Fresh embryo transfer was the most common treatment used in our study sample. When we restricted the comparison to fresh transfer versus naturally conceived, the number of significant CpGs increased from 607 to 800, suggesting that associations were attenuated when all the ART procedures were combined into a single group. Three of these CpGs were also significant when the smaller group conceived by frozen embryo

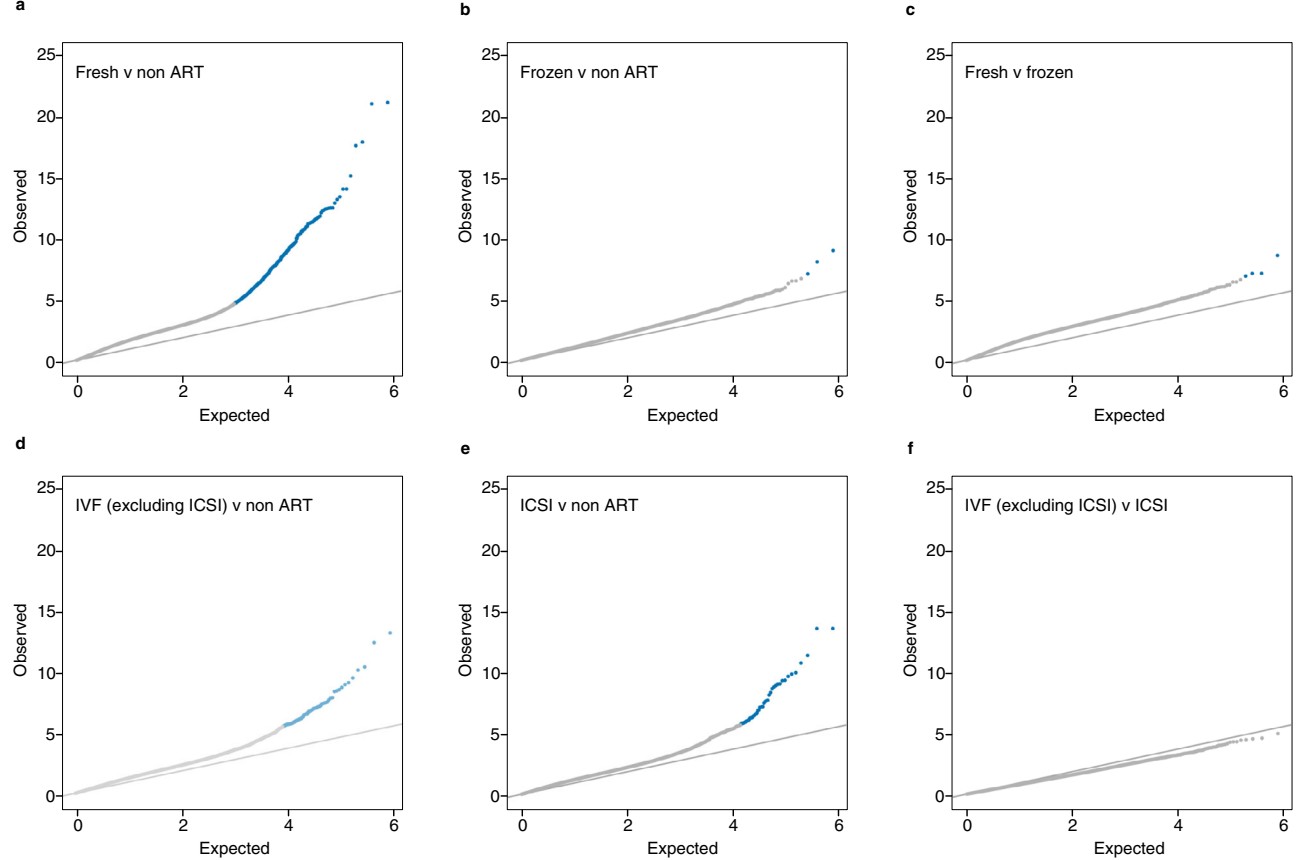

**Fig. 5 Quantile-Quantile plots of observed versus expected $-\log_{10} P$ values for methylation differences with different ART procedures.** The observed $-\log_{10} P$ values on the y-axis (calculated from $t$ statistics) were derived from two-sided $t$ tests comparing the DNA methylation levels between: **a** ART conceived newborns using fresh embryo transfer ($n = 764$) and naturally conceived newborns ($n = 983$), **b** ART conceived newborns using frozen embryo transfer ($n = 126$) and naturally conceived newborns ($n = 983$), **c** ART conceived newborns using frozen embryo transfer ($n = 126$) and fresh embryo transfer ($n = 764$), **d** ART conceived newborns using in vitro fertilization (IVF) without intracytoplasmic sperm injection (ICSI) ($n = 511$) and naturally conceived newborns ($n = 983$), **e** ART conceived newborns using IVF with ICSI ($n = 359$) and naturally conceived newborns ($n = 983$), **f** ART conceived newborns using IVF with ICSI ($n = 359$) and ART conceived newborns using IVF without ICSI ($n = 511$). The adjusting variables were maternal age, maternal smoking, maternal body mass index, child sex, parity, and plate ID. Blue dots indicate significant CpGs at FDR < 0.01, while gray dots indicate nonsignificant CpGs. The diagonal lines in the panels (**a–f**) indicate the expected $-\log_{10} P$ values if there were no difference between the groups. Source data are provided as a Source Data file.

transfer was compared to the naturally conceived group. There were three other significant CpGs when comparing frozen embryo transfer to fresh transfer. Together, these findings suggest that methylation differences are found with both frozen and fresh transfer, and that methylation levels may differ between frozen and fresh embryo treatment. Previous studies have reported differences in birth outcomes between frozen and fresh transfer, especially in relation to birthweight[4]. Freezing could impact fetal DNA methylation. However, it is possible that changes in the intrauterine environment due to maternal hormonal stimulation prior to fresh embryo transfer could influence birth outcomes. We did not find any significant differences between newborns conceived with ICSI versus those conceived without ICSI, suggesting that other aspects of ART are more important in our findings.

It is plausible that the media used to culture embryos before insertion or freezing may impact epigenetic stability and the epigenetic reprogramming[33]. Duration of incubation before embryo insertion may also play a role. Our study consisted of children conceived at different fertility clinics across Norway from 2000 to 2008, and we were not able to examine differences in growth medium or incubation time between the clinics.

Parents who use ART to conceive differ from those who conceive without ART in several ways[34]. In contrast to previous studies[12–16], we were able to control for potential confounders that have been associated with both the use of ART and with DNA methylation in newborns, such as maternal age[35], maternal smoking[36], and maternal BMI[37]. However, as information on some of these potential confounders and other key variables was derived from questionnaires, potential misclassification may have limited our ability to fully account for confounders. Time to pregnancy (TTP) is difficult to capture precisely, although long and short TTPs are more reliably recalled[38]. Misclassifications may weaken an association and reduce the ability to detect differences with TTP. Although women included in MoBa are not completely representative of all Norwegian mothers[39], the biological implications are expected to be similar for all women and children. This is supported by a study showing that associations between pregnancy exposures and child outcomes were not biased by selection into the MoBa study[39].

Although most of the significant CpGs had higher methylation, there was a global shift to lower methylation in ART conceived newborns. It is possible that some aspects of the ART procedure (e.g., culture media) lead to this reduced DNA methylation.

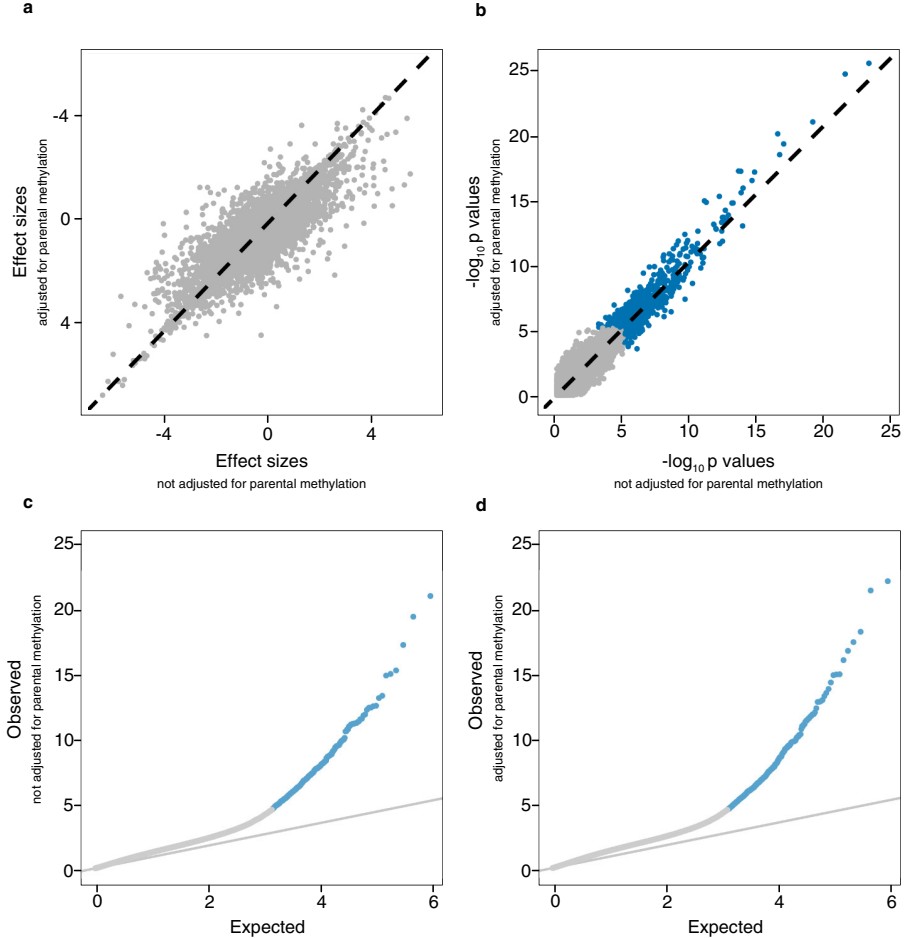

**Fig. 6 Comparison of the size of effects and *P* values with and without adjustment for parental methylation.** The effect sizes and observed $-\log_{10} P$ values (calculated from *t* statistics) were derived from two-sided *t* tests for comparing the DNA methylation levels between ART conceived newborns ($n = 962$) and naturally conceived newborns ($n = 983$). **a** The effect sizes on the *x*-axis were adjusted for maternal age, maternal smoking, maternal BMI before pregnancy, parity, child sex, and plate number, and the effect sizes on the *y*-axis were adjusted for all the covariates listed and parental methylation. The Pearson's correlation coefficient was 0.95. **b** The observed $-\log_{10} P$ values were calculated from the effect sizes in panel (**a**) and their standard errors. Blue dots indicate the significant CpG sites at FDR < 0.01, while gray dots indicate nonsignificant CpG sites. **c** The expected $-\log_{10} P$ values adjusted for maternal age, maternal smoking, maternal BMI before pregnancy, parity, child sex, and plate number are on the *x*-axis, and the observed *P* values are on the *y*-axis. Blue dots indicate the significant CpG sites at FDR < 0.01, while gray dots indicate nonsignificant CpG sites. The slope in gray indicates the null distribution of no differences between ART and naturally conceived. **d** The expected $-\log_{10} P$ values adjusted for maternal age, maternal smoking, maternal BMI before pregnancy, parity, child sex, plate number, and parental DNA methylation are on the *x*-axis, and the observed *P* values are on the *y*-axis. Blue dots indicate the significant CpG sites at FDR < 0.01, while gray dots indicate nonsignificant CpG sites. The slope in gray indicates the null distribution of no differences between ART and naturally conceived. Source data are provided as a Source Data file.

However, the implications of this observation are uncertain, and whether it persists after birth would be interesting to explore at older ages.

Taken together, our results support the hypothesis that ART-induced DNA methylation alterations may impact health and disease risk later in life. Further research is needed to clarify whether epigenetic differences observed at birth persist into adulthood and contribute to differences in health-related outcomes between ART-conceived and naturally conceived children. Since ART was introduced in the late 1970s, potential long-term consequences in adulthood have not yet been addressed. Longitudinal assessments of DNA methylation and outcomes are required to investigate whether DNA methylation patterns established in early life and potential alterations in ART-conceived children persist and influence future health outcomes. If future studies show that DNA methylation patterns contribute to more adverse health outcomes in ART-conceived

children and in adulthood, research on possible targeted interventions and improvements of ART would be urgently needed.

Our results support the hypothesis that ART procedures influence DNA methylation in fetal life. The epigenetic differences between ART and naturally conceived children were not explained by parental DNA methylation or parental subfertility. Longitudinal assessments of DNA methylation are necessary to establish whether ART-induced differences in DNA methylation persist and influence later health outcomes.

## Methods
This study was approved by the Regional Committees for Medical and Health Research Ethics of South East Norway (#2017/1362). All participants provided written informed consent. The establishment of MoBa and initial data collection were based on a license from the Norwegian Data Protection Agency and an approval from the Regional Committees for Medical and Health Research Ethics. The MoBa cohort is now regulated by the Norwegian Health Registry Act.

**Table 2 Number of differentially methylated CpGs in 176 genes with significant differences between ART conceived and naturally conceived newborns.**

| No. of differentially methylated CpGs per gene | No. of genes | % of genes | Gene(s) |
|---|---|---|---|
| 1 | 121 | 68.8 | ADO, AFAP1, AFDN-DT, AFMID, AGBL4, AKR1B1, ALK, ANKRD26, ANKRD30A, APBA1, APOA5, ARHGAP22, ARL4A, ATP8B3, BANP, BET1, BOLA1, CACNA1H, CCSAP, COX16, CRACDL, CYB5B, EBF3, EMCN, EML1, ENPP6, ERLIN2, FANCC, FCHO1, FGF22, FIGNL1, FSCN2, FUCA1, GABRB3, GALNT9, GGN, GOPC, GPATCH1, GRB2, GTF2A1L:STON1-GTF2A1L, HOXD4, INPP5A, KCNK5, KLHL8, LBP, LILRB4, LIPE-AS1, LOC102467224, LOC440434, LOC644669, LONP2, LTN1, MAL2, MFSD14C, MIR646HG, MIR7-3, MRPL3, MRPS25, MSX1, MTOR, MYH7B, MYO1D, NABP2, NBEAL1, NDUFAF1, NPEPPS, NUP205, NUPL1, NUS1, OCA2, OR4C13, OR51A2, OR5D16, OVCH1-AS1, PACS2, PCSK7, PDE11A, PDE7B, PIM1, POLR1E, PPP1R13L, PRRG2;NOSIP, PTGR2, PTPRQ, ANKRD20A19P, RASL11B, RBPJL, RESP18, RGR, RHPN2, RPS6KA2, RPS6KC1, RPS6KC2, RYR2, SCARA5, SCG2, SLC17A7, SLC5A1, SLC9A2, SLC9B1, SMYD3, SNAP91, SNORD115-45, SPESP1, SRP19, SVIL, TBC1D12, TBC1D15, TBC1D19, TCTN1, TEX14, TOGARAM1, TSPEAR, TTC29, TXNDC12, UQCRB, USP48, WDR19, ZNF708, ZNF709, ZPBP |
| 2 | 20 | 11.4 | ATG4C, ATG5, COL9A1, DRC1, LOXL1, LUC7L, MATN4, MTERF, NLRP2, NTNG2, OXCT1, PRR23A, PRR23B, RAD54B, RFPL2, RWDD2B, TCERG1L, TSTD1, VPS51, YIF1B |
| 3 | 11 | 6.3 | ANKRD2, CDH20, EIF2B3, FAM228A, MORC2-AS1, PRSS41, RNF126P1, RPS2P32, SDHAF1, STX1, USH1G |
| 4 | 4 | 2.3 | GLI2, KLK4, SLC35F3, TMEM263 |
| 5 | 3 | 1.7 | LDHAL6A, NECAB3, PNLDC1 |
| 6 | 7 | 4.0 | APC2, C17orf98, CHRNE, GET1, MAN2B1, SNCB, SYCP1 |
| 7 | 2 | 1.1 | PRR25, ZNF727 |
| 8 | 6 | 3.4 | CCDC166, LOC650226, MTNR1B, PIWIL1, PXDNL, RBM46 |
| 9 | 0 | 0.0 | |
| 10 | 1 | 0.6 | BRCA1 |
| 11 | 1 | 0.6 | HLA-DQB2 |

**Study population**. The Norwegian Mother, Father and Child Cohort study (MoBa)[25] is an ongoing cohort study with more than 110,000 children and their parents. Women were recruited during pregnancy between 1999 and 2008, and approximately 40% of those invited participated. Partners were included from 2001. Participants are followed with questionnaires and by linkage to the Medical Birth Registry of Norway (MBRN). Blood samples were collected from the parents during pregnancy, and from the umbilical cord at delivery[25,40]. We randomly selected 992 mother–father–child trios with naturally conceived children and 978 trios with ART conceived children who met the following criteria: the children were born in years 2001–2009, they were singletons with a birth record, mothers had returned the first MoBa questionnaire around week 18 of pregnancy, and DNA samples from complete trios were available.

**DNA methylation measurements and data processing**. DNA samples were shipped to the Institute of Life & Brain Sciences at the University of Bonn in Germany for processing and DNA methylation measurements using the Illumina Infinium MethylationEPIC BeadChip array[41]. The samples were bisulfite converted using the EZ-96DNA methylation-Lightning™MagPrep kit (Zymo Research, Irvine, USA). Raw data for the Illumina MethylationEPIC array was obtained using Genome Studio 2011.2. After receiving the iDAT files, we performed quality control using the RnBeads (v2.2.0) R (v3.5.0) package[42]. We removed 44,210 cross-hybridizing probes[43] and 16,117 additional probes where the last three bases overlapped with a SNP. Probes with a high detection $p$ value (>0.01) were also removed. In addition, the different batches were subjected to the greedycut algorithm to remove samples and probes with outlying DNA methylation patterns. The remaining DNA methylation signals were corrected for background noise using the normalizing function ENmix.oob[44]. The signal intensity of all the samples was visually inspected using the output of control probes from RnBeads. Whenever a CpG site was removed from one batch due to poor quality or high detection $p$ value, it was subsequently removed from all remaining batches. This resulted in 770,586 autosomal CpGs in the final set for the current analyses. The Beta-mixture quantile normalization (BMIQ)[45] in the watermelon (v1.26.0) R (v3.5.0) package[46] was used to normalize type I and type II probes. During quality control, we excluded two children with empty plate wells, one child with outlier values, three children with corrupt images, and 19 children with high background signals, leaving 1945 children, 1956 mothers, and 1949 fathers (1917 complete trios) for analysis.

**ART and subfertility**. Fertility clinics mandatorily report ART to the birth registry with details on ART procedures, such as in vitro fertilization (IVF) with or without intracytoplasmic sperm injection (ICSI), and fresh or frozen-thawed embryo transfer. ART was defined as "any ART" (excluding inseminations) and coded as a binary

variable, and further divided into combinations of subtypes. When information on the ART procedure was unavailable ($n = 79$), it was coded as "unknown procedure". Subfertility among those who conceived naturally was defined according to the number of months it took to conceive (time to pregnancy; TTP), from the MoBa questionnaire at 18 weeks of gestation. TTP was categorized as <3 months, 3–8 months, 9–12 months, and >12 months.

**Covariates**. MoBa questionnaires provided information on educational level, height and weight, and maternal and paternal smoking during pregnancy. Information on child sex, parity, and gestational age was obtained from the birth registry. Gestational age at birth was based on ultrasound measurements when available ($n = 1,835$), and when not, it was based on the last menstrual period for naturally conceived children ($n = 24$) and the time of embryo insertion for ART conceived children ($n = 82$). Child sex can be incorrectly recorded in the birth registry. Based on their DNA methylation profile, we reclassified one female as male, and five males as females.

**Statistics**

*DNA methylation differences in ART versus naturally conceived newborns*. Linear mixed model regression was used to compare associations between DNA methylation at each CpG in ART and naturally conceived newborns. For all statistical analyses, beta values were transformed to M-values. A few maternal characteristics related to the use of ART have been associated with cord blood DNA methylation, such as age[35], smoking[36], and body mass index (BMI)[37]. We adjusted for these characteristics, together with child sex (strongly associated with DNA methylation) and parity (strongly associated with ART). All samples were randomly placed on plates before measuring DNA methylation to reduce batch effects. However, we also included plate ID as a random effect to further correct for batch effects. In sensitivity analyses we included additional adjustments for gestational age at birth, birthweight, maternal educational level, paternal age, and cell-type composition in cord blood. All models included only persons with information on the variables in the respective models (see Table 1 for information on missing data). Two-sided Wald tests were used to assess the significance of adjusted associations between ART and DNA methylation in each model. The Benjamini-Hochberg procedure[47] was used to control for multiple testing by applying a false discovery rate (FDR) cut-off of <0.01. We calculated the two first principal components (PC) of the 607 significant CpGs found in the initial EWAS. These two PCs were plotted for ART-conceived and naturally conceived girls and boys separately and for mothers and fathers. The PCs showed distinct clustering in the children, which was not observed in the parents. In additional analyses, we compared DNA methylation in the following groups of newborns: fresh embryo transfer versus naturally conceived

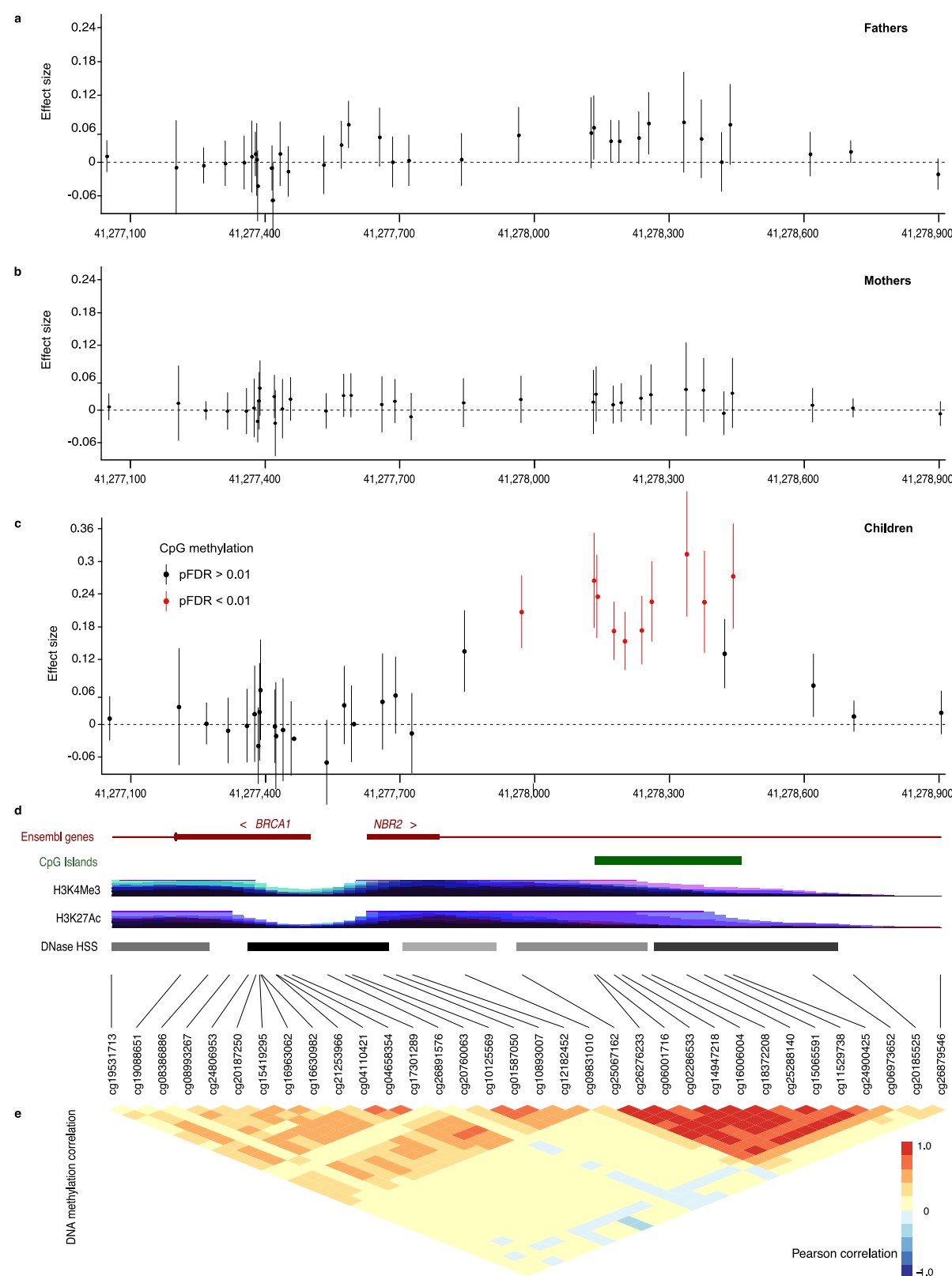

newborns, frozen embryo transfer versus naturally conceived, fresh embryo transfer versus frozen embryo transfer, and IVF with ICSI versus naturally conceived, IVF without ICSI versus naturally conceived, and IVF with ICSI compared directly to IVF without ICSI.

*Underlying parental subfertiliy.* To address whether DNA methylation differences in the newborns were due to inherited variation related to parental subfertility, we adjusted for parental DNA methylation at each CpG site. In addition, we analyzed DNA methylation in naturally conceived newborns using time to pregnancy (TTP) as a measure of parental fertility. A linear mixed model was fitted to investigate the association between DNA methylation in the newborns with TTP as an ordinal variable, adjusting for maternal age, maternal smoking, maternal BMI, child sex, parity, and plate ID. Further, we examined the differences in the two most contrasting groups: TTP < 3 months to TTP > 12 months.

**Fig. 7 Differences in DNA methylation between fathers and mothers of naturally conceived and ART-conceived children at the *BRCA1-NBR2* locus.** In panels (**a**–**c**), each dot represents the coefficient from the regression of DNA methylation on ART-conceived vs naturally conceived newborns, and the vertical bar represents the 95% confidence interval. The region shown is chr17:41277059-41278906 (hg19). **a** 969 fathers for ART-conceived children, 980 fathers for naturally conceived children were examined. The adjusting variables were paternal age, paternal smoking, and paternal BMI. **b** 970 mothers for ART conceived children, 986 mothers for naturally conceived children were examined. The adjusting variables were maternal age, maternal smoking, maternal BMI before pregnancy, and parity. **c** 962 ART conceived and 983 naturally conceived newborns were examined. The adjusting variables were maternal age, maternal smoking, maternal BMI before pregnancy, child sex, parity, and plate ID. **d** The *BRCA1-NBR2* bidirectional promoter showing transcription start and orientation, CpG island (green) and epigenetic marks (purple) representing the site of the promoter (H3K4me3, active promoter; H3K27Ac, active regulatory element; DNA hypersensitive sites (grayscale) open chromatin). **e** The heatmap shows Pearson's correlation coefficients of the DNA methylation across the 34 CpGs within the interval. Source data are provided as a Source Data file.

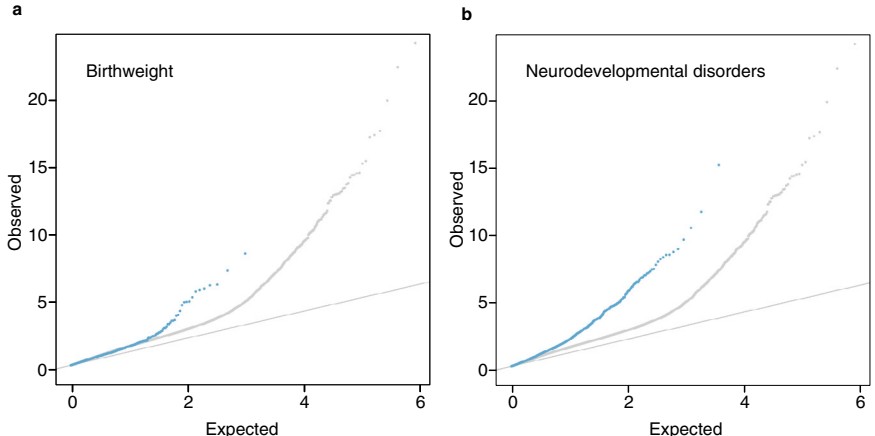

**Fig. 8 Quantile-Quantile plots of observed versus predicted −log$_{10}$ P values for methylation differences with ART in CpGs associated with birthweight and neurodevelopment.** The observed −log$_{10}$ P values on the y-axis (calculated from *t* statistics) were derived from two-sided *t* tests for: (**a**) the association between DNA methylation levels and birthweight on a continuous scale (in grams) and (**b**) the difference in DNA methylation levels between subjects with Mendelian neurodevelopmental disorders and controls. Blue dots indicate the P values for a subset of CpGs that have been reported to be significant for birthweight and neurodevelopmental disorders. For comparison, the P values for the overall associations of ART compared to naturally conceived are shown in gray. The diagonal lines indicate expected P values if there were no difference between the groups. Source data are provided as a Source Data file.

*Implications for health outcomes.* We examined potential health implication of genes with multiple significant CpGs. We also compared our results with genes reported in two recently published EWASs that used the EPIC array[12,14]. ART has consistently been associated with birthweight[32], and several studies suggest associations between ART and neurodevelopment[4,5]. A recent meta-analysis of 8,825 newborns identified 914 CpGs with significant DNA methylation differences associated with birthweight[26]. Another study identified 34 CpGs as differentially methylated in a study of 42 Mendelian neurodevelopmental disorders[27]. We explored whether these previously identified CpGs were overrepresented among the CpGs differentially methylated in ART conceived newborns.

Analyses were performed using Stata version 16 (StataCorp) and R (www.r-project.org; v3.5.0). The linear mixed models were implemented using the *rint.reg* function in the R package Rfast (v1.9.2).

**Reporting summary**. Further information on research design is available in the Nature Research Reporting Summary linked to this article.

## Data availability

CpGs were annotated to include further information. The initial annotation was obtained from the Illumina EPIC manifest file (https://support.illumina.com/array/array_kits/infinium-methylationepic-beadchip-kit/downloads.html), which provides information on CpG probe ID, probe sequence, chromosome position, gene name etc. Approved gene names were obtained from HUGO Gene Nomenclature Committee (HGNC; https://www.genenames.org). In order to provide stable gene IDs, ENSEMBL IDs (www.ensembl.org) are included. Information on human genetic disorders was obtained from Online Mendelian Inheritance in Man (OMIM; https://www.omim.org). Information on mouse mutation phenotypes was obtained from Mouse Genome Informatics (MGI; http://www.informatics.jax.org).

The data that support the findings of this study are available from the Norwegian Institute of Public Health (NIPH), but restrictions apply regarding the availability of these data, which were originally used under specific approvals for the current study and are therefore not publicly available. The individual level data are available under restricted access due to regulations and access can only be given after approval by the Norwegian Ethical committees under the provision that the applications are consistent with the consent provided. Access can be obtained by application to the Norwegian Institute of Public Health using a form available on the English language portion of its website at https://www.fhi.no/en/studies/moba/. Specific questions regarding access for data in this study can be directed to Siri.Haberg@fhi.no. The data generated in this study are provided in the Supplementary Information. Source data are provided with this paper.

## Code availability

R scripts are available from the authors upon request.

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

## Acknowledgements

The Norwegian Mother, Father, and Child Cohort Study is supported by the Norwegian Ministry of Health and Care Services and the Ministry of Education and Research. We are grateful to all the participating families who take part in this ongoing cohort study. This work was funded by the Research Council of Norway through its Centres of Excellence funding scheme, project number 262700 (S.E.H., C.M.P., Y.L., R.L., M.C.M., H.N., E.Ø.C., W.R.P.D., A.J., J.B., K.L.H., H.K.G., P.M.) and by the Norwegian Institute of Public Health (S.E.H., A.J., J.B., H.K.G., P.M.).

## Author contributions

Author contributions: S.E.H., H.K.G., R.L., C.M.P. designed research; C.M.P., M.C.M., H.N., Y.L., R.L. conducted analyses, S.E.H., C.M.P., R.L., Y.L., M.C.M., H.N., E.Ø.C., W.R.P.D., A.J., J.B., K.L.H., H.K.G., P.M. performed interpretation of data, S.E.H., C.M.P., R.L., M.C.M., E.Ø.C., W.R.P.D., H.N., A.J. drafted the paper, J.B., K.L.H., H.K.G., P.M., S.E.H., E.Ø.C., A.J., R.L. conducted review and revisions; S.E.H., H.K.G., P.M. provided funding acquisition, project administration, and resources. All authors approved the submitted version and have agreed both to be personally accountable for the author's own contributions and to ensure that questions related to the accuracy or integrity of any part of the work, even ones in which the author was not personally involved, are appropriately investigated, resolved, and the resolution documented in the literature.

## Competing interests

The authors declare no competing interests. The Funding agency had no role in the conceptualization, design, data collection, analysis, decision to publish, or preparation of the manuscript.
