## [Peer Review File · Nature Communications]

DNA methylation in newborns conceived by assisted reproductive technologyREVIEWER COMMENTS

Reviewer #1 (Remarks to the Author):

Important and compelling work that deserves attention from ART field.

Reviewer #2 (Remarks to the Author):

This is an exciting and very important EWAS aimed at identifying DNA methylation variation in cord blood of pregnancies conceived by assisted reproductive technologies. There are several strengths of the study, including;

- Very well written and overall well presented
- Large sample size enriched for ART - This is by far the largest study of its kind yet carried out with cord blood from nearly 1000 ART conceived and 1000 naturally conceived pregnancies. Data on ART subtype are also available
- Methylation data from parents – Such trios are rare and provide a unique strength to the study, allowing an examination of heritability of methylation of interest. Unfortunately, this aspect of the study is not exploited to its full potential
- Sample sourced from a single well-characterised cohort- Although several other studies have attempted to build sufficient power, these have combined data from multiple (primarily observational) cohorts with previously existing methylation array data and often poorly characterised ART pregnancies
- Analysis is of high quality with appropriate adjustments – continues the high quality analyses from this team
- Global methylation analysis findings are provocative and would be strengthened by some supporting figures and additional data presented in results
- ART subtype analysis QQ plots – showing enrichment with some procedures relative to others (novel)
- BW and Neurodev QQ plots - showing enrichment for probes linked to these phenotypes (novel)

Areas that could be strengthened

- Results as a whole are very brief and somewhat cursory
 - o A total of 607 CpGs were identified as differentially methylated in ART vs non-ART individuals at birth that were not explained by parental methylation status. Some PCA or heatmap analysis of these probes across cord blood and parents for both groups would enhance this finding
 - o Figure 1 OK
 - o Figure 2 would be better if comparing ART relative to non-ART trios rather than just ART. This should demonstrate the hypomethylation in ART offspring
 - o Figure 3 is OK but it would be a valuable addition to showing the distribution of methylation at specific DMRs for some of the highest ranked genes/regions, particularly relative to those previously identified/published
 - o Figure 4 and Figure 5 would also benefit from having some examples of differential methylation at specific genes/sites
 - o Some examples showing ART specific methylation relative to non-art offspring and parents generally, would have been an interesting addition
 - o Supp table 2 and supp table 3 should include effect size data rather than just effect direction. They should also list individual CpG probe names in all instances even if this does take up additional space.

Reviewer #3 (Remarks to the Author):

This study is unique because of the large number of ART individuals and matched controls and is therefore an important source of replication of previous findings and identification of novel ART-associated DNA methylation variation. The analysis is well described and tables (e.g. supp table 2) are useful for the reader and nicely summarize the data. I have specific comments to improve the relationship between the results and discussion (for example some findings are never discussed). Further, I felt that in some cases the authors asked an interesting question, but then did not utilize the full potential of the data to give a satisfactory answer (e.g. the fresh v frozen embryo transfer section).

Major

1. Analysis in Figure 2 is an interesting idea.

a. However, I am curious how many decimals the authors use to determine this 75% hypomethylation figure, and what it might possibly mean to have such a pattern. Without a p-value significance cut-off, I'd be very hesitant to conclude that the ART individuals are globally hypomethylated – a feature of cancer genomes.

b. Second, was this calculated on mean DNA methylation of the groups? Is it possible to determine for each sample in each group whether the individual is more or less methylated than any other individual? As in "x% of ART individuals show hypomethylation compared to the mean of all individuals, as opposed to y% of naturally conceived...."

c. Finally, this finding is not discussed further and in fact the word 'hypomethylated' is not repeated again after this paragraph. My suggestion is to either remove this part or see if you can pinpoint where this hypomethylation is occurring (e.g. repetitive elements or globally, or at specific genomic contexts?). Another aspect that makes me confused about the message of this analysis is that right after we learn that 65% of DMPs are hypermethylated in ART.

2. Regarding the fresh v frozen analysis. This analysis is interesting, and the authors are right to use their large sample size to ask this important question. I would be interested to see the direction of DNA methylation change at these loci and to see what the frozen group looks like at fresh-embryo specific DMPs. Specifically, is it simply a matter of more samples = more power = more DMPs at FDR <0.01? It could be that the frozen group shows the same direction and degree of change, but just doesn't reach the FDR cut-off. If in fact, there are probes that only change in fresh with no difference at all in frozen, then those are interesting probes to report.

3. In the abstract, "Both fresh and frozen embryo transfer showed DNA methylation differences" is an interesting conclusion based on this finding (page 6, line 110): "When frozen embryo transfer (n=125) was compared to naturally conceived, there were three statistically significant CpGs (Fig. 4b)." I assume that the authors come to this conclusion based on the original analysis that includes all ART subtypes (both fresh and frozen). Then why perform the technique-specific analysis (fresh only vs natural...) at all if it is not summarized in the abstract? My suggestion is to dive deeper into this finding and identify any DMPs that are clearly only present in fresh transfer, as this would suggest hormones are playing a part at these loci.

4. Table 2: how far away are the DMPs from genes? How were probes linked to genes? By distance or by Illumina annotation?

5. Regarding ICSI vs IVF. Why was there no analysis of ICSI v natural and IVF v natural, similarly to what was performed for fresh transfer v natural?

Minor

6. Consider revising the location of the commas, or the whole sentence, in beginning of results: "After data processing and quality control, DNA methylation data generated using the Illumina MethylationEPIC array, was available for 962 ART conceived and 983 naturally conceived"

7. Supplementary table 4 is referenced in the text first. Perhaps this table should then be Supp Table 1.

RESPONSE TO REVIEWER COMMENTS

Reviewer #1 (Remarks to the Author):

Important and compelling work that deserves attention from ART field.

Reviewer #2 (Remarks to the Author):

This is an exciting and very important EWAS aimed at identifying DNA methylation variation in cord blood of pregnancies conceived by assisted reproductive technologies. There are several strengths of the study, including;

- Very well written an overall well presented
- Large sample size enriched for ART - This is by far the largest study of its kind yet carried out with cord blood from nearly 1000 ART conceived and 1000 naturally conceived pregnancies. Data on ART subtype are also available
- Methylation data from parents – Such trios are rare and provide a unique strength to the study, allowing an examination of heritability of methylation of interest. Unfortunately, this aspect of the study is not exploited to its full potential
- Sample sourced from a single well-characterised cohort- Although several other studies have attempted to build sufficient power, these have combined data from multiple (primarily observational) cohorts with previously existing methylation array data and often poorly characterised ART pregnancies
- Analysis is of high quality with appropriate adjustments – continues the high quality analyses from this team
- Global methylation analysis findings are provocative and would be strengthened by some supporting figures and additional data presented in results
- ART subtype analysis QQ plots – showing enrichment with some procedures relative to others (novel)
- BW and Neurodev QQ plots - showing enrichment for probes linked to these phenotypes (novel)

Areas that could be strengthened

- Results as a whole are very brief and somewhat cursory
 - o A total of 607 CpGs were identified as differentially methylated in ART vs non-ART individuals at birth that were not explained by parental methylation status. Some PCA or heatmap analysis of these probes across cord blood and parents for both groups would enhance this finding

Response: We have added a supplementary figure plotting PCA1 and PCA2 for the 607 significant CpGs for ART in girls and boys separately, and for the mothers and fathers (now Supplementary Figure 1). This shows that for the 607 significant CpGs, the distribution of PCAs in parents using ART overlaps with non-ART parents, while the distribution for the 607 top hits is shifted in ART newborns compared to non-ART newborns. These results have been added to the Results section and described in more details in the Methods section.

o Figure 1 OK

o Figure 2 would be better if comparing ART relative to non-ART trios rather than just ART. This should demonstrate the hypomethylation in ART offspring

Response: We have now modified the title and legend of Figure 2 to clarify that the effect sizes shown are from the comparison of ART to non-ART, and that non-ART is the reference group. In the legend for Figure 2, we added that the effect sizes were from the regression of DNA methylation on ART versus non-ART. We also clarified the main finding of these analyses in the Results section and moved the median values of the distribution with confidence intervals from the figure footnote to the Results section.

o Figure 3 is OK but it would be a valuable addition to showing the distribution of methylation at specific DMRs for some of the highest ranked genes/regions, particularly relative to those previously identified/published

Response: We have added a supplementary figure (Supplementary Fig. 6) showing the distribution of methylation for the eight genes with the highest number (eight or more) of differentially methylated CpGs between ART and naturally conceived newborns (see Table 2 for the CpG count for these genes).

o Figure 4 and Figure 5 would also benefit from having some examples of differential methylation at specific genes/sites

Response: Figure 4 now includes all comparisons of frozen, fresh, IVF with ICSI, and IVF without ICSI. We have added a new table with a list of genes with significant CpGs overlapping between fresh and frozen comparisons, and genes with significantly differentially methylated CpGs for the direct comparison of frozen to fresh (Supplement Table 2).

o Some examples showing ART specific methylation relative to non-art offspring and parents generally, would have been an interesting addition

Response: We have added Supplementary Figure 7 plotting methylation across the *BRCA1/NBR2* locus in both the children and parents, showing that methylation is altered in ART children but not the parents. We have now included sentences that describe this finding in both the Results and Discussion.

o Supp table 2 and supp table 3 should include effect size data rather than just effect direction. They should also list individual CpG probe names in all instances even if this does take up additional space.

Response: Effect sizes and probe names are now added to these Supplementary Tables. For our study and for that of Yeung et al., we provide effect sizes from the statistical modelling. For the Novakovic et al. study, effect sizes were not provided in their results, so we provide methylation differences between ART and controls. The direction of the effect sizes were consistent across the studies.

Reviewer #3 (Remarks to the Author):

This study is unique because of the large number of ART individuals and matched controls and is

therefore an important source of replication of previous findings and identification of novel ART-associated DNA methylation variation. The analysis is well described and tables (e.g. supp table 2) are useful for the reader and nicely summarize the data. I have specific comments to improve the relationship between the results and discussion (for example some findings are never discussed). Further, I felt that in some cases the authors asked an interesting question, but then did not utilize the full potential of the data to give a satisfactory answer (e.g. the fresh v frozen embryo transfer section).

Major

1. Analysis in Figure 2 is an interesting idea.

a. However, I am curious how many decimals the authors use to determine this 75% hypomethylation figure, and what it might possibly mean to have such a pattern. Without a p-value significance cut-off, I'd be very hesitant to conclude that the ART individuals are globally hypomethylated – a feature of cancer genomes.

Response: Figure 2 displays an overall pattern of lower methylation in ART conceived children when compared to naturally conceived. This is an intriguing finding, however, we do not know what the potential implications are, if any. The median values of the effect sizes were statistically significant (we provide the confidence intervals) in the ART to non-ART comparison. This could be due to the large number of CpGs in this comparison, although the differences in methylation are small for most CpGs. The main message here is that the distribution is shifted towards negative values, and more importantly, that this shift is not seen when comparing ART and non-ART parents. We have made the title and legend clearer in Figure 2 and explained how the effect sizes were obtained (regression coefficients of DNA-methylation of ART versus non-ART), and that these effect sizes were obtained for more than 770 000 CpGs. This information is also included in the Results section. In the Discussion, we clarified the main finding and its unknown consequences, without further speculation.

b. Second, was this calculated on mean DNA methylation of the groups? Is it possible to determine for each sample in each group whether the individual is more or less methylated than any other individual? As in $\hat{x}\%$ of ART individuals show hypomethylation compared to the mean of all individuals, as opposed to $y\%$ of naturally conceived....”

Response: Figure 2 represents the distribution of the regression coefficients for all CpGs on the form of $CpG = b \cdot ART + \text{confounders}$, that is, the distribution of all the different values of b . Since ART is a binary variable, this would reflect an adjusted average difference, after accounting for the effect of any confounders. What we see is that the coefficient tends to be more negative for children than for parents, indicating a loss of methylation after ART, that is the adjusted relationship $CpG = b \cdot ART + \text{confounders}$ are negative. Unfortunately, we cannot estimate the % with hypomethylated ART individuals.

c. Finally, this finding is not discussed further and in fact the word hypomethylated is not repeat again after this paragraph. My suggestion is to either remove this part or see if you can pinpoint where this hypomethylation is occurring (e.g. repetitive elements or globally, or at specific genomic contexts?). Another aspect that makes me confused about the message of this analysis is that right after we learn that 65% of DMPs are hypermethylated in ART.

Response: We have added a paragraph in the Discussion to highlight the contrast between the overall lower methylation and the top hits which mostly were more methylated. We also clearly state in the Discussion that we do not know what the possible effect of this hypomethylation could be.

2. Regarding the fresh v frozen analysis. This analysis is interesting, and the authors are right to use their large sample size to ask this important question. I would be interested to see the direction of DNA methylation change at these loci and to see what the frozen group looks like at fresh-embryo specific DMPs. Specifically, is it simply a matter of more samples = more power = more DMPs at FDR <0.01? It could be that the frozen group shows the same direction and degree of change, but just doesn't reach the FDR cut-off. If in fact, there are probes that only change in fresh with no difference at all in frozen, then those are interesting probes to report.

Response: We agree that it is important to address any potential difference between frozen vs fresh embryo transfer. We have added a new table (Supplementary Table 2) with a list of CpGs that were significant CpGs in the comparison of fresh versus frozen and the overlapping CpGs that were significant in both frozen and fresh transfer when compared to naturally conceived. There were three differentially methylated CpGs in the comparison of frozen to non-ART. However, these were also significant in fresh vs non-ART, indicating that these CpGs are associated with the ART procedure itself, irrespective of whether fresh or frozen embryo transfer is used. However, when comparing frozen to fresh, three other CpGs were significant. Moreover, the number of CpGs increased when frozen was excluded from the analyses. Together these results indicate that there may be differences in methylation between fresh and frozen. This is now clarified in the Discussion.

3. In the abstract, "Both fresh and frozen embryo transfer showed DNA methylation differences" is an interesting conclusion based on this finding (page 6, line 110): "When frozen embryo transfer (n=125) was compared to naturally conceived, there were three statistically significant CpGs (Fig. 4b)." I assume that the authors come to this conclusion based on the original analysis that includes all ART subtypes (both fresh and frozen). Then why perform the technique-specific analysis (fresh only vs natural...) at all if it is not summarized in the abstract? My suggestion is to dive deeper into this finding and identify any DMPs that are clearly only present in fresh transfer, as this would suggest hormones are playing a part at these loci.

Response: Due to more children with fresh transfers than frozen in our study sample, we have substantially larger power to detect significant hits in the fresh vs natural, than in the frozen vs natural. When comparing fresh directly to frozen, three CpGs were significantly different (FDR<0.01, Supplementary Table 2). One was not annotated to any gene, and the other two were located near known genes (*KLC1* and *ASB6*). The implication for this in ART is unknown. We have expanded the Discussion regarding the findings between fresh and frozen (see also our previous response to point 2 above).

4. Table 2: how far away are the DMPs from genes? How were probes linked to genes? By distance or by Illumina annotation?

Response: The CpGs were linked to genes based on the Illumina Manifest file. This is described in the Methods section in the paragraph labeled "CpG annotation".

5. Regarding ICSI vs IVF. Why was there no analysis of ICSI v natural and IVF v natural, similarly to what was performed for fresh transfer v natural?

Response: We agree that all comparisons should be presented, and have now added the ICSI vs non-ART and the IVF versus non-ART in Figure 4. We have listed the number of significant CpGs for each method, and the overlap between them in the Results.

Minor

6. Consider revising the location of the commas, or the whole sentence, in beginning of results:

“After data processing and quality control, DNA methylation data generated using the Illumina MethylationEPIC array, was available for 962 ART conceived and 983 naturally conceived”

Response: Thank you for pointing this out, the sentence has been revised.

7. Supplementary table 4 is referenced in the text first. Perhaps this table should then be Supp Table 1.

Response: Due to its size, we have moved Supplementary Table 4 to a separate supplement (Supplement 1). The old Supplemental Table 4 is now labeled Supplement Table 1. The other supplementary tables and figures are now in Supplement 2.

REVIEWERS' COMMENTS

Reviewer #2 (Remarks to the Author):

Overall, this is now a much improved manuscript. The majority of my concerns have been addressed satisfactorily. However, the finding of global hypomethylation in association with ART remains one of considerable significance and it would be beneficial to deal with this a little more thoroughly if possible, particularly in relation to identifying any key characteristics of the 607 probes highlighted.

- Inclusion of PCA in supp Fig 1 strengthens the findings of a global shift in DNA methylation. Figure 2 is similarly compelling, but it would have been interesting to present corresponding figures for subsets of probes according to genomic feature (eg. CGI, shores, shelves, gene associated, regulatory elements, encode chromatin features etc.)
- It would be nice to see how these 607 probes behave if another age matched dataset is combined with the current set to exclude any age related effect, though this isn't needed in the current ms.
- Supp Fig 1 and Supp Fig 3 are interesting, far more so than current figures 4-6 in the main text. Similarly, Supp Fig 7 is a really nice representation of the specificity of the observed effect to ART and would be ideal as a figure in the manuscript if the parents were combined or only one shown.

Reviewer #3 (Remarks to the Author):

This is a fine manuscript. However, the hypomethylation finding in Figure 2 is still of concern, because there is no context for it. Can the authors show what this exact figure would look like if a cancer sample is compared to a healthy tissue? I.e. How does the 'hypomethylation' in ART compare to actual global hypomethylation in cancer. Most results for the search 'hypomethylation' lead to cancer papers, so we should be careful of using this word, knowing full well that ART conceived individuals read these papers with interest. Perhaps, 'genome-wide slightly lower methylation' is more accurate, as first, the changes are small, second the EPIC array is not designed to look at 'global' methylation. It does not cover repetitive elements, partially methylated domains, etc.

REVIEWERS' COMMENTS

Reviewer #2 (Remarks to the Author):

Overall, this is now a much improved manuscript. The majority of my concerns have been addressed satisfactorily. However, the finding of global hypomethylation in association with ART remains one of considerable significance and it would be beneficial to deal with this a little more thoroughly if possible, particularly in relation to identifying any key characteristics of the 607 probes highlighted.

- Inclusion of PCA in supp Fig 1 strengthens the findings of a global shift in DNA methylation. Figure 2 is similarly compelling, but it would have been interesting to present corresponding figures for subsets of probes according to genomic feature (eg. CGI, shores, shelves, gene associated, regulatory elements, encode chromatin features etc.)

Response: We have added a new figure (now Supplementary Fig. 1) where we show the overall methylation distribution according to genomic features.

- It would be nice to see how these 607 probes behave if another age matched dataset is combined with the current set to exclude any age related effect, though this isn't needed in the current ms.

Response: We agree, and we look forward to future studies on this topic. We do not know of any similar datasets currently available. Our findings are on the children, and results were robust for adjustment for gestational age. All results are also adjusted for parental age, so we believe age related effects are mainly accounted for.

- Supp Fig 1 and Supp Fig 3 are interesting, far more so than current figures 4-6 in the main text. Similarly, Supp Fig 7 is a really nice representation of the specificity of the observed effect to ART and would be ideal as a figure in the manuscript if the parents were combined or only one shown.

Response: We have moved the suggested figures into the main text (Supp Fig 1 is now panel c in Fig 3, Supp Fig 3 is now Fig.4, and Supp Fig. 7 is now Fig. 7 in main document).

Reviewer #3 (Remarks to the Author):

This is a fine manuscript. However, the hypomethylation finding in Figure 2 is still of concern, because there is no context for it. Can the authors show what this exact figure would look like if a cancer sample is compared to a healthy tissue? I.e. How does the 'hypomethylation' in ART compare to actual global hypomethylation in cancer.

Response: Showing figures on cancer tissue hypomethylation is outside the scope of our paper. In addition, given the very large number of cancer types (tissues, stages etc.), we do not think a meaningful comparison could be made here. We have been careful to not interpret or speculate in potential health effects (if any) of the global shift in methylation.

Most results for the search 'hypomethylation' lead to cancer papers, so we should be careful of using this word, knowing full well that ART conceived individuals read these papers with interest. Perhaps, 'genome-wide slightly lower methylation' is more accurate, as first, the changes are small, second the EPIC array is not designed to look at 'global' methylation. It does not cover repetitive elements, partially methylated domains, etc.

Response: We have changed the wording as suggested: First sentence of these section of results have been changed to:

“Newborns conceived by ART exhibited genome-wide slightly lower methylation (Fig. 2a).”